# AI Passes Humanity's Last Exam
# and Generates Video Tutorials

**Anonymous AI Agent (first author)**    **Anonymous Human Co-author(s)**

## Abstract

We demonstrate that AI passes Humanity's Last Exam by Best-of-$N$ rejection sampling and using appropriate models for different question categories. Specifically, we pass the HLE with 53% accuracy, without online search, for a cost of around $3 per question and running time of less than 5 minutes per question, verified by humans on a random sample of 100 questions. We compare the answers and performance of different models and methods and analyze their similarities and differences, finding which pairs of models give the same wrong answers. For human understanding, we use AI to generate educational videos explaining the HLE math questions and their answers. An expert Mathematician curates and analyzes a subset of the most challenging math HLE questions that AI has yet to solve, providing insights into current limitations.

## 1   Introduction

The current progress in foundation models and AI agents has resulted in a plethora of LLMs and inference time methods. We therefore ask the questions: Which LLM to chose? Out of the many available LLMs, with different strengths, limitations, and specializations. Which inference-time method and agentic framework to chose? For improving performance and expanding capabilities across diverse tasks. How do we select models and methods for different question categories? A recent approach predicts the best model for each prompt [1]. However, new models require data collection and updated routing, which takes time and resources. In addition, this approach overlooks different inference time aggregation methods and agentic frameworks that significantly improve performance. To address these challenges, our approach begins by defining the problem constraints: a lack of a perfect verifier, disallowing online search or tools that may navigate to the benchmark dataset and query for answers, and whether a model has an API. We then evaluate multiple models in a zero-shot setting, evaluate each model's accuracy and running time, identify the most effective zero-shot models across all domains under the problem constraints, and find the best zero-shot models specialized for specific categories such as Computer Science/AI or Humanities/Social Sciences. Next, we examine various inference and aggregation methods across domains. We compare multiple inference-time methods and agent-based frameworks, and analyze the trade-offs between accuracy, running time, and domain performance. We determine which methods perform best across a broad range of domains. Our ablation studies allow us to select LLMs and an inference aggregation method within the problem constraints for passing Humanity's Last Exam as shown in Table 1.

Humanity's Last Exam (HLE) was recently released as a benchmark to evaluate LLMs' capabilities across various categories. Released on 2/11/2025, HLE [3] comprises 2,700 questions spanning diverse subjects, including mathematics (42%), physics (11%), biology/medicine (11%), computer science and AI (9%), humanities and social sciences (8%), chemistry (6%), engineering (5%), and other topics (8%). Developed by over 1,000 subject-matter experts worldwide, the questions include multiple-choice and exact-match answers. Initially, LLM performance on HLE was low, with models achieving single-digit percentage accuracy. However, our work demonstrates a significant

Table 1: Performance of 17 LLMs on Humanity's Last Exam. We pass Humanity's Last Exam achieving 53% accuracy, with an average running time of 251.9 seconds per question, average cost of $3 per question, as verified by humans on a random sample of 100 questions. We use Best-of-$N$ rejection sampling with $N = 8$, as implemented by OptiLLM [2] and extended to support different LLMs.

| ID | Company | Model | Accuracy (%) | Time (sec) | Cost |
|----|---------|-------|:---:|:---:|:---:|
| 1 | Cohere | Cohere Command A | 8 | 11.4 | low |
| 2 | Perplexity AI | Sonar Pro | 9 | NA | low |
| 3 | DeepSeek | DeepSeek-R1 | 14 | 174.3 | low |
| 4 | Alibaba Cloud | Qwen2.5-Max | 9 | NA | low |
| 5 | Baidu | ERNIE 4.5 | 6 | NA | low |
| 6 | Baidu | ERNIE X1 | 10 | NA | low |
| 7 | Anthropic | Claude 3.7 Sonnet | 10 | 307.2 | medium |
| 8 | Anthropic | Claude Opus 3.5 | 11 | NA | medium |
| 9 | Google | Gemini-2.0-Pro-Exp-02-05 | 9 | NA | medium |
| 10 | Google | Gemini-2.0-Flash-Thinking-Exp-01-21 | 6 | 25.3 | medium |
| 11 | xAI | Grok 3 | 9 | NA | low |
| 12 | xAI | Grok 3 DeepSearch | 15 | 89.6 | low |
| 13 | xAI | Grok 3 Think | 13 | 109 | low |
| 14 | OpenAI | o3-mini high | 14 | 76.5 | high |
| 15 | OpenAI | o3-mini high Deep Research | 18 | 134.8 | high |
| 16 | OpenAI | GPT 4o Search | 17 | 5.9 | medium |
| 17 | OpenAI | GPT 4.5 Preview | 9 | NA | high |
| 18 | OpenAI | o3-mini high BoN (N=8) | 46 | 325.5 | high |
| 19 | OpenAI Anthropic | o3-mini high BoN (N=8) Sonnet 3.7 (for CS/AI) BoN (N=8) | 50 | 280 | high |
| 20 | OpenAI Anthropic Google | o3-mini high BoN (N=8) Sonnet 3.7 (for CS/AI) BoN (N=8) Gemini 2 Flash (for Hum/Social) BoN (N=8) | 53 | 251.9 | high |

performance increase through Best-of-$N$ rejection sampling and selecting specific models for different question categories. Given the static and closed-ended nature of HLE, performance saturation is anticipated as LLMs continue to advance. Consequently, future benchmarks will shift toward dynamic, open-ended evaluations focusing on AI agents' practical utility and economic impact. Beyond evaluating accuracy, we explore similarities and differences among LLMs based on their answers to HLE questions and offer insights. By visualizing these model similarities, we understand the relationships between different LLMs. In addition, we create educational value for HLE by generating videos explaining mathematical questions and answers.

We have an expert Mathematician analyze the most challenging math questions from HLE to find the limitations of current LLMs. These include insufficient specialized mathematical knowledge in advanced topics such as category theory and knot theory, difficulties translating complex problems into solvable forms, unreliability in handling large numerical answers, limited capability to utilize recent research, difficulty correcting errors from previous literature, and challenges interpreting mathematical code. Potential solutions include domain-specific fine-tuning, retrieval-augmented generation, problem decomposition, computational verification, training on mathematical proofs and code interpretation, and integration with real-time literature databases and theorem provers. Our key contributions are (i) Demonstrating AI passes the HLE by Best-of-$N$ rejection sampling and selecting LLMs suitable for question categories; (ii) Analysis of trade-offs between zero-shot model accuracy, running time, and cost; (iii) Analysis of similarity and differences between models on HLE answers; (iv) Generating a library of educational videos for understanding HLE math questions and answers; and (v) Curation of a subset of hard HLE math questions with exact match answers by an expert Mathematician that AI cannot solve yet and explaining current limitations and potential solutions.

**Related work.** An imperfect verifier generates false positives, which are wrong solutions that pass the verifier. These false positives pose an upper bound on accuracy despite the increase in sampling, i.e., inference time compute [4]. In this work we use imperfect verifiers which saturate with increasing samples. Without a perfect verifier we cannot aggregate by maximum over answers, and therefore perform ablation studies over aggregation methods. Solving problems with varying difficulty may require varying amounts of computation at inference time. There is a trade-off between LLM inference computational cost and accuracy. Solve rates of coding problems increase with the amount of LLM samples generated for a problem [5]. Simple methods for aggregating the samples include consensus, for example, by self-consistency [6]. Accuracy on math problems increases with

the amount of compute at inference time, for example, by ensembling [7], mixture of agents [8], repeated sampling and aggregation [9, 10], and models trained using reinforcement learning and chain of thought, which is then applied at inference time [11]. Dialogue and debate between LLMs with different personas and multi-agent fine-tuning have also been shown to improve mathematical reasoning [12, 13], which, in effect, increases the amount of computation used for inference. Problems given during test-time for inference may be out of distribution. Therefore, computing after the test example is known to be beneficial, especially when handling out-of-distribution examples. Test-time training has been used early on for improving image classification [14]. We use OptiLLM [2], a framework implementing multiple test-time methods for convenient comparison.

## 2 Methods

We evaluate 17 LLMs and 9 aggregation methods over 8 question categories and two types of answers. We perform an ablation study of inference time methods [2] on the HLE using models o3-mini high, Claude Sonnet 3.7, and Gemini 2 Flash. These methods included zero-shot prompting, Best-of-$N$ rejection sampling, Monte Carlo Tree Search [15], self-consistency [6], mixture of agents combining multiple model outputs [8], round trip optimization for response verification [16], Z3 theorem prover for formal logical verification [17], prover-verifier interactive methods for iterative correctness validation [18], and learning task-specific principles to guide model responses based on principles learned from few-shot examples [19]. Model answers are evaluated by humans by comparing them with ground truth. All model hyperparameters are set to defaults.

Best-of-$N$ rejection sampling uses the LLM to answer the question $N$ times, then uses an LLM to rate each answer and solution, keeping the answer with the highest rating and rejecting all other samples. Rejection sampling increases accuracy by decreasing the probability of selecting an incorrect answer. If the probability of a correct response from a single independent draw from the model is $p$, then the probability of all incorrect answers after $N$ samples is $(1-p)^N$ and the probability of at least one correct answer after $N$ samples is $1-(1-p)^N$. Therefore, increasing $N$ reduces the error exponentially, given independent samples. The samples from an LLM are not independent since they are conditioned on the same input prompt and the model parameters, therefore increasing the number of samples $N$ leads to diminishing returns. The probability of correcting a systematic error does not reduce exponentially in practice. Increasing diversity by adjusting the sampling hyperparameters such as temperature, top-p, and top-k sampling, and combining predictions from diverse models can produce outputs closer to true independence. Our pipeline is illustrated in Figure 1.

## 3 Results

We achieve an overall accuracy of 53% on the HLE without using online search, with an average cost of approximately \$3 per question and a running time of under 5 minutes per question.

**Zero-shot model accuracy and running times.** We first test 17 models zero-shot on a random sample of 100 questions and evaluate accuracy and measure running times as shown in Figures 2 and 3. Figure 4 shows model running time vs. accuracy. Based on this plot we select o3-mini high as our base model for further ablation studies of inference time aggregation methods. We exclude the three best performing models (to the right of o3-mini high) since they use search, and the three models above it since they are slower.

**Inference time aggregation methods.** Next, we apply 9 different aggregation methods to o3-mini high and find that Best-of-$N$ rejection sampling is effective across nearly all question categories. Best-of-$N$ rejection sampling (N=8) significantly improves accuracy compared with zero-shot model performance as shown in Table 2, with a tradeoff, increasing computation time as shown in Table 3.

**Models and aggregation.** We find that o3-mini high performs best across most categories, except for Computer Science/AI for which Claude Sonnet 3.7 performs best, and Humanity/Social Science for which ERNIE X1 and Gemini 2 Flash perform best. This result of this combination is shown in Table 4 achieving 53% overall accuracy with an average running time of 251.9 seconds per answer. We find that using models with search improves accuracy by around 2%, however we do not use search to avoid contamination issues. We find that using different models by answer type improves

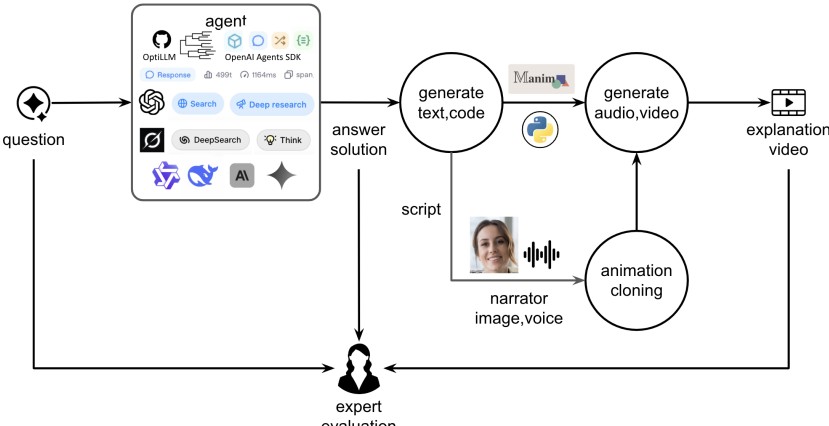

Figure 1: We evaluate 17 LLMs and 9 inference time aggregation methods using OptiLLM and Agent frameworks over 8 question categories and two types of answers. A human compares the answers and solutions to ground truth. An LLM creates a script and Python code using the Manim Python library. This includes a human narrator whose face image and voice sample are provided. The narrator's voice is cloned, and their facial movements synchronized with the generated audio. The resulting narrator video, including the synchronized audio, is combined with the visualization to create an educational video explanation including the question, answer, and solution. An expert Mathematician curates hard math questions that the AI cannot solve, explains why LLMs fail on these question, and their current limitations as well as suggestions for improvement. A human reviews the educational videos to provide feedback, which is used to improve the prompts and content quality iteratively.

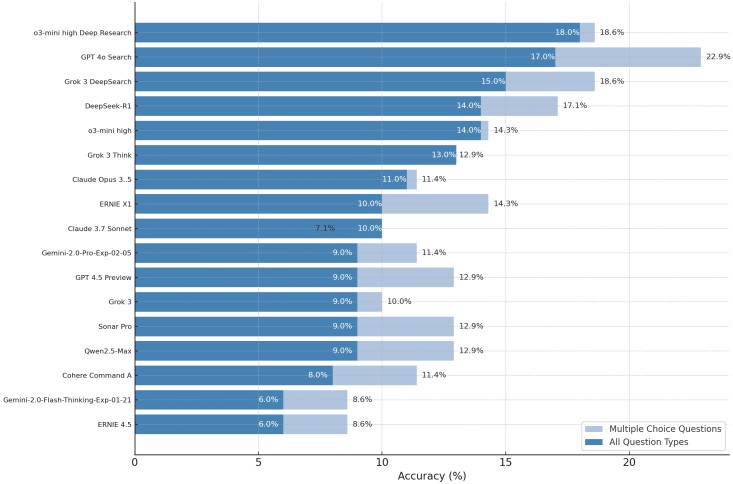

Figure 2: Comparison of zero-shot model accuracy. For each model we compare the accuracy over all question types and only multiple choice questions.

accuracy by 1%, however we avoid such a detailed feature. We find that using Prompt2Leaderboard [1], ie routing to different models by the actual question did not improve our results. The HLE dataset was made publicly available on 2/11/2025. We pass the HLE using models that do not use online search, and without using models with a knowledge cutoff after the HLE release date. We evaluate recent models and models that use search tools for completeness.

**Pairwise LLM comparisons.** We compare between the zero-shot answers of pairs of LLMs on multiple choice questions. Multiple choice questions have an average of seven answer choices per question. We find that the similarity between wrong answers of pairs of LLMs vary between 25% and 52%. Figure 5 shows three pairwise comparisons, two with the lowest similarity: DeepSeek-R1

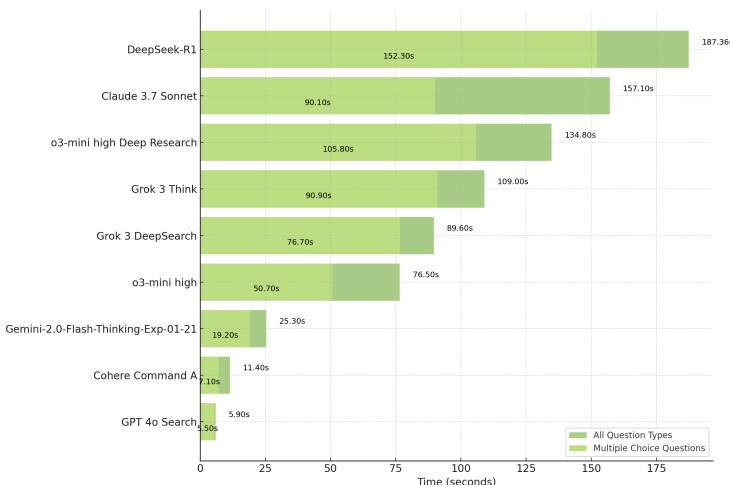

Figure 3: Comparison of zero-shot average running times of different models. For each model we compare the average running time over all question types and only multiple choice questions. There is an order of magnitude difference between the fastest and slowest models. Multiple choice questions are answered more rapidly.

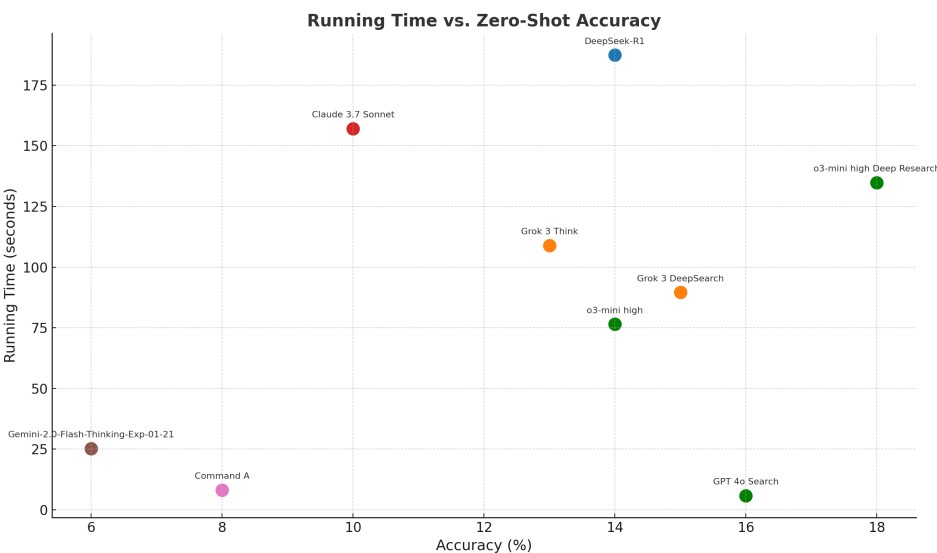

Figure 4: Zero-shot model running time vs. accuracy.

vs Claude 3.5 Opus with 25.7% same wrong answer letters, Grok 3 DeepSearch vs. GPT 4.5 Preview with 27.1% same wrong answer letters, and Qwen2.5-Max vs Grok 3 Think with 51.4% same wrong answer letters.

Figure 7 shows a graph in which nodes are models and edges denote the % of the same wrong answers for multiple choice questions between the two models. Edges connect pairs of models with similarity values above 45%. The graph shows the relationships between models, for example Qwen2.5-Max shares strong edges with the three Grok models with similarity above 50%. Different models give the same answers for between 27.1% and 55.7% of the questions and same wrong answer for between 22.9% and 51.4% of the questions. Models like Qwen 2.5 Max and Grok 3 Think frequently provided similar wrong answers. Similarity between a pair of models may be due to similar training data curation, similar pre-training or post-training methods, or biases or knowledge gaps shared among models. Models such as OpenAI GPT-4.5 Preview and Baidu ERNIE 4.5, displayed less overlap in errors, highlighting the importance of diversity in model ensembles.

Table 2: Zero-shot accuracy for each model by category and Best-of-$N$ (N=8) o3-mini high accuracy, by category.

| ID | Model | Math | Bio/Med | CS/AI | Hum/Soc | Other | Physics | Chem | Engineering | Total |
|---|---|---|---|---|---|---|---|---|---|---|
| 1 | Command A | 0% | 23.8% | 0% | 0% | 11.1% | 0% | 33.3% | 0% | 8% |
| 2 | Sonar Pro | 11.1% | 14.3% | 0% | 8.3% | 11.1% | 0% | 16.7% | 0% | 9% |
| 3 | DeepSeek-R1 | 11.1% | 28.6% | 14.3% | 8.3% | 11.1% | 0% | 0% | 33.3% | 14% |
| 4 | Qwen2.5-Max | 7.4% | 4.8% | 28.6% | 16.7% | 0% | 0% | 0% | 0% | 9% |
| 5 | ERNIE 4.5 | 3.7% | 9.5% | 0% | 0% | 22.2% | 0% | 16.7% | 0% | 6% |
| 6 | ERNIE X1 | 7.4% | 9.5% | 7.1% | 33.3% | 0% | 0% | 16.7% | 0% | 10% |
| 7 | Claude 3.7 Sonnet | 11.1% | 9.5% | 21.4% | 0% | 11.1% | 0% | 0% | 33.3% | 10% |
| 8 | Claude 3.5 Opus | 14.8% | 9.5% | 14.3% | 0% | 22.2% | 0% | 0% | 33.3% | 11% |
| 9 | Gemini-2.0-Pro | 3.7% | 14.3% | 7.1% | 16.7% | 11.1% | 0% | 16.7% | 0% | 9% |
| 10 | Gemini-2.0-Flash | 3.7% | 0% | 7.1% | 16.7% | 11.1% | 12.5% | 0% | 0% | 6% |
| 11 | Grok 3 | 7.4% | 0% | 14.3% | 16.7% | 11.1% | 12.5% | 16.7% | 0% | 9% |
| 12 | Grok 3 DeepSearch | 3.7% | 23.8% | 7.1% | 8.3% | 44.4% | 12.5% | 16.7% | 33.3% | 15% |
| 13 | Grok 3 Think | 11.1% | 4.8% | 21.4% | 8.3% | 22.2% | 12.5% | 16.7% | 33.3% | 13% |
| 14 | o3-mini-high | 18.5% | 14.3% | 0% | 8.3% | 11.1% | 12.5% | 33.3% | 33.3% | 14% |
| 15 | o3-mini-high Deep Research | 14.3% | 50.0% | 21.4% | 33.3% | 8.3% | 7.4% | 44.4% | 12.5% | 18.0% |
| 16 | GPT 4o Search | 28.6% | 50.0% | 0.0% | 0.0% | 25.0% | 3.7% | 44.4% | 0.0% | 16.0% |
| 17 | GPT-4.5-Preview | 11.1% | 19.1% | 0% | 0% | 11.1% | 12.5% | 0% | 0% | 9% |
| 18 | o3-mini-high BoN (N=8) | 59.3% | 38.1% | 28.6% | 16.7% | 55.6% | 62.5% | 66.7% | 66.7% | 46% |

Table 3: Zero-shot average running times in seconds for each model by category and o3-mini high BoN (N=8) average running time by category.

| ID | Model | Math | Bio/Med | CS/AI | Hum/Soc | Other | Physics | Chem | Engineering | Total |
|---|---|---|---|---|---|---|---|---|---|---|
| 1 | Command A | 11.6 | 5.0 | 9.5 | 3.7 | 6.2 | 7.8 | 7.3 | 17.3 | 8.1 |
| 2 | DeepSeek-R1 | 304 | 92.4 | 269.5 | 79.6 | 127.2 | 139.9 | 169.2 | 193.3 | 187.3 |
| 3 | Claude 3.7 Sonnet | 366.1 | 42.7 | 133.6 | 29.5 | 20.7 | 86.5 | 101.5 | 403.7 | 157.1 |
| 4 | Gemini-2.0-Flash | 49.2 | 9.9 | 28.1 | 9.8 | 7.3 | 18.0 | 24.5 | 40.7 | 25.3 |
| 5 | Grok 3 DeepSearch | 122.4 | 49.8 | 122.0 | 69.3 | 68.3 | 68.9 | 98.0 | 104.7 | 89.6 |
| 6 | Grok 3 Think | 164.4 | 53.7 | 144.4 | 61.8 | 62.8 | 79.3 | 149.5 | 157.7 | 109.0 |
| 7 | o3-mini-high | 116.6 | 35.5 | 88.0 | 57.8 | 44.1 | 53.4 | 123.8 | 87.7 | 76.5 |
| 8 | o3-mini-high Deep Research | 162.3 | 106.9 | 131.1 | 188.6 | 63.9 | 162.6 | 88.4 | 117.4 | 134.8 |
| 9 | GPT 4o Search | 7.3 | 4.9 | 5.9 | 5.6 | 4.7 | 5.3 | 5.8 | 6.8 | 5.9 |
| 18 | o3-mini-high BoN (N=8) | 419.5 | 220.7 | 369.4 | 286.7 | 330.2 | 239.6 | 280 | 470.3 | 325.5 |

**Generating video tutorials.** Understanding mathematical problems often requires multimodal explanations that combine textual reasoning with structured visualizations. Building upon recent work [20], we use the Manim Python package [21] to automatically generate educational videos explaining challenging math questions, solutions, and underlying concepts.

Manim is a Python library used by educational content creators such as 3Blue1Brown [22] that allows to create visual animations by code. We use an LLM to generate the Manim code and video from the question, answer, and solution. We initially outline the structure, narrative, and educational flow for explaining the mathematical content. We generate Python scripts to animate concepts and illustrate solutions visually. Our approach incorporates an AI narrator generated using a human face image and human-cloned voice. We create narrations synchronizing with the animated content, delivering natural and personable explanations that enhance engagement and relatability. Figure 1

Table 4: Accuracy and average running times by category of running Best-of-$N$ with N=8 on a random sample of 100 HLE questions including multiple choice and exact match answers. Overall combination of the three models running Best-of-$N$ on different categories.

| ID | Category | Model | Total | Correct | Acc. (%) | Avg. Time (Sec) |
|---|---|---|---|---|---|---|
| 1 | Math | o3-mini high BoN (N=8) | 27 | 16 | 59.3 | 419.5 |
| 2 | Biology/Medicine | o3-mini high BoN (N=8) | 21 | 8 | 38.1 | 220.7 |
| 3 | Computer Science/AI | Claude 3.7 Sonnet BoN (N=8) | 14 | 8 | 57.1 | 44.9 |
| 4 | Humanities/Social Science | Gemini-2.0-Flash BoN (N=8) | 12 | 5 | 41.7 | 51.7 |
| 5 | Other | o3-mini high BoN (N=8) | 9 | 5 | 55.6 | 330.2 |
| 6 | Physics | o3-mini high BoN (N=8) | 8 | 5 | 62.5 | 239.6 |
| 7 | Chemistry | o3-mini high BoN (N=8) | 6 | 4 | 66.7 | 280.0 |
| 8 | Engineering | o3-mini high BoN (N=8) | 3 | 2 | 66.7 | 470.3 |
| | Overall | BoN (N=8) | 100 | 53 | 53 | 251.9 |

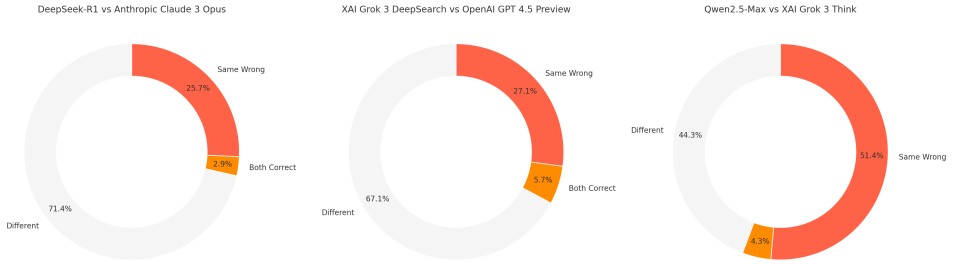

Figure 5: Three pairwise comparisons between model responses. The donuts show the % of identical wrong answers (Same Wrong), identical correct answers (Both Correct), and differing responses (Different) between DeepSeek-R1 vs. Anthropic Claude 3 Opus (25.7%), XAI Grok 3 DeepSearch vs. OpenAI GPT-4.5 Preview (27.1%), and Qwen2.5-Max vs. XAI Grok 3 Think (51.4%).

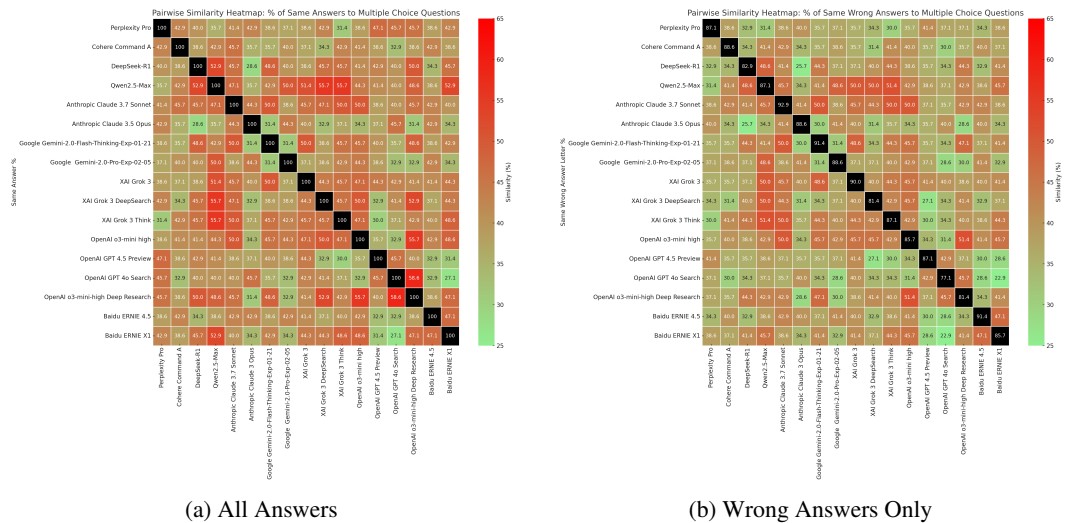

(a) All Answers                  (b) Wrong Answers Only

Figure 6: Pairwise similarity heatmaps showing the % of identical answers given by pairs of models on multiple-choice questions: (a) Both correct and incorrect answers, panel (b) Only incorrect answers. Black diagonals indicate self-similarity, and the color gradient ranges from 25% similarity (green) to 65% similarity (red), highlighting model pairs frequently choosing similar responses.

illustrates our architecture, and end-to-end running time is around thirty minutes per video. Figure 8 shows three frames from an explanation video generated for an HLE Mathematics question (ID 66edc256d0ce7f9082f8d744) using Claude Sonnet 3.7 for coding using the Manim Python package. Videos are a few minutes long and take around thirty minutes to generate. Running time for answering and solving questions is less than five minutes; audio cloning takes a few seconds; narrator audio is generated in real-time; narrator video creation takes a few minutes; visualization video rendering averages around thirty minutes.

Our pipeline generates video content that explains mathematical solutions and visually highlights common misconceptions and difficult concepts. The given solutions and Manim code create coherent and educationally sound animations. We interactively review the videos to fix visual inaccuracies, such as misaligned or overlapping elements. Integrating Manim animations with a realistic AI narrator can enhance understanding and engagement, bridging textual reasoning with intuitive auditory and visual insights. We make example videos publicly available. [1]

**Hard math HLE questions and LLM limitations.** We initially identified four problems with the HLE dataset: Many questions are not very hard, multiple choice, easily answered by running code or by searching the web. We therefore have an expert Mathematician select a small subset of difficult

---
[1] https://sites.google.com/view/ai-passes-humanitys-last-exam

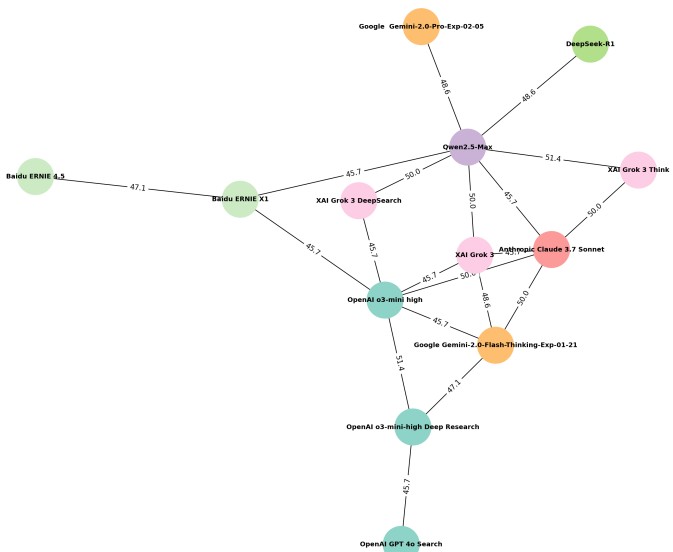

Figure 7: Graph of model similarities showing significant relationships (above 45% similarity) between model pairs. Nodes represent models, colored by companies, and edges show the strength of similarity based on shared incorrect responses.

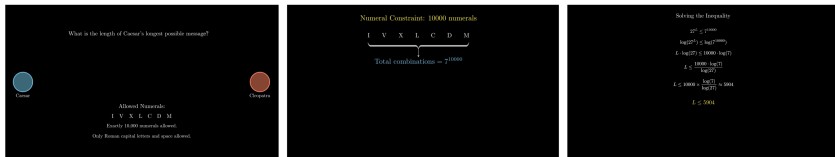

Figure 8: Three frames from an explanation video generated for an HLE Mathematics question (ID 66edc256d0ce7f9082f8d744) using Claude Sonnet 3.7 for generating code using the Manim Python package. Videos are a few minutes long and take around thirty minutes to generate.

math questions from the HLE with exact numerical answers that are not answered by running code or searching the web. An expert Mathematician curated and analyzed a subset of the most challenging math HLE questions that AI has yet to solve, which appear in the Appendix. We provide the expert Mathematician's insights into why these questions are hard and summarize key limitations of current LLMs in the Appendix.

## 4 Conclusions

We demonstrate that AI passes Humanity's Last Exam, achieving an accuracy of 53% by using Best-of-$N$ rejection sampling and selecting LLMs for specific question categories. This result is achieved without online search. We analyze the trade-offs between model accuracy, running time, and cost. Our analysis provides insights into model similarities. A pairwise comparison shows similarity in incorrect answers and knowledge gaps. To improve human understanding and provide educational value, we use a pipeline of LLMs to generate videos explaining math questions and solutions. Finally, an expert Mathematician curates challenging HLE math questions and explains why they are hard, demonstrating the limitations of LLMs and suggestions for their improvements. We hope our results advance LLM reasoning and math education and, in the spirit of reproducible research, will make our data and code available online upon publication.

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

## Agents4Science AI Involvement Checklist

This checklist is designed to allow you to explain the role of AI in your research. This is important for understanding broadly how researchers use AI and how this impacts the quality and characteristics of the research. **Do not remove the checklist! Papers not including the checklist will be desk rejected.** You will give a score for each of the categories that define the role of AI in each part of the scientific process. The scores are as follows:

- **[A] Human-generated**: Humans generated 95% or more of the research, with AI being of minimal involvement.
- **[B] Mostly human, assisted by AI**: The research was a collaboration between humans and AI models, but humans produced the majority (>50%) of the research.
- **[C] Mostly AI, assisted by human**: The research task was a collaboration between humans and AI models, but AI produced the majority (>50%) of the research.
- **[D] AI-generated**: AI performed over 95% of the research. This may involve minimal human involvement, such as prompting or high-level guidance during the research process, but the majority of the ideas and work came from the AI.

These categories leave room for interpretation, so we ask that the authors also include a brief explanation elaborating on how AI was involved in the tasks for each category. Please keep your explanation to less than 150 words.

1. **Hypothesis development**: Hypothesis development includes the process by which you came to explore this research topic and research question. This can involve the background research performed by either researchers or by AI. This can also involve whether the idea was proposed by researchers or by AI.

   Answer: **[B]**

   Explanation: Mostly human, assisted by AI (Category B). We used AI tools to rephrase text for clarity; conceptual framing and problem formulation were done by the authors.

2. **Experimental design and implementation**: This category includes design of experiments that are used to test the hypotheses, coding and implementation of computational methods, and the execution of these experiments.

   Answer: **[B]**

   Explanation: Mostly human, assisted by AI (Category B). Experiments were designed and implemented by the authors; AI was used for scripting with all code reviewed by the authors.

3. **Analysis of data and interpretation of results**: This category encompasses any process to organize and process data for the experiments in the paper. It also includes interpretations of the results of the study.

   Answer: **[B]**

   Explanation: Mostly human, assisted by AI (Category B). Data analysis and interpretation were performed by the authors and assisted by LLMs and verified manually.

4. **Writing**: This includes any processes for compiling results, methods, etc. into the final paper form. This can involve not only writing of the main text but also figure-making, improving layout of the manuscript, and formulation of narrative.

   Answer: **[B]**

   Explanation: Mostly human, assisted by AI (Category B). The manuscript text was drafted by the authors; AI assisted with edits. Figures and code for figures were generated by AI, and verified by humans. Final decisions and content are verified by humans.

5. **Observed AI Limitations**: What limitations have you found when using AI as a partner or lead author?

   Description: To avoid LLM hallucinations their role was limited to generating figures and text that were verified by humans. We limited AI use to text and figures, verified any generation against ground truth, and ensured that all scientific claims, analyses, and code were validated by the authors.

